# NXP081, DNA Aptamer–Vitamin C Complex Ameliorates DNFB-Induced Atopic Dermatitis in Balb/c Mice

**DOI:** 10.3390/nu15194172

**Published:** 2023-09-27

**Authors:** Sanggon Lee, Hyun-Jong Ahn, Yong Seek Park, Ji-Hyun Kim, Yoon-Seong Kim, Jeong-Je Cho, Cheung-Seog Park

**Affiliations:** 1Department of Microbiology, College of Medicine, Kyung Hee University, Seoul 02447, Republic of Korea; sg2307@khu.ac.kr (S.L.); ahnh@khu.ac.kr (H.-J.A.); yongseek@khu.ac.kr (Y.S.P.); jjcho@khu.ac.kr (J.-J.C.); 2Nexmos Co., Ltd., U-Tower, 767, Sinsu-ro, Yongin-si 16827, Republic of Korea; helen@nexmos.com (J.-H.K.); yk525@rwjms.rutgers.edu (Y.-S.K.)

**Keywords:** atopic dermatitis, vitamin C, DNA aptamer

## Abstract

Atopic dermatitis (AD) is a chronic inflammatory disease characterized by dry and itchy skin. Recently, it has been reported that oxidative stress is involved in skin diseases, possibly including AD. Vitamin C, also referred to as ascorbic acid, is a vital water-soluble compound that functions as an essential nutrient. It plays a significant role as both an antioxidant and an additive in various pharmaceutical and food products. Despite the fact that vitamin C is easily oxidized, we have developed NXP081, a single-stranded DNA aptamer that selectively binds to vitamin C, thereby inhibiting its oxidation. The objective of the current research was to examine the impact of NXP081, an animal model of AD induced by 2,4-dinitrofluorobenzene (DNFB). The experimental drug NXP081, when taken orally, showed promising results in reducing inflammation and improving the skin conditions caused by DNFB. The administration of NXP081 resulted in a significant reduction in ear swelling and a noticeable improvement in the appearance of skin lesions. In addition, the administration of NXP081 resulted in a significant decrease in the migration of mast cells in the skin lesions induced by DNFB. Moreover, NXP081 inhibited the production of interferon-gamma (IFN-γ) in CD4^+^ T cells that were activated and derived from the lymph nodes. Our findings provide useful information about the anti-inflammatory effect of NXP081 on AD.

## 1. Introduction

Atopic dermatitis (AD), or atopic eczema, is a highly prevalent skin condition that leads to inflammation. About 10 to 20 out of 100 children in developing countries have it, and approximately 1 to 3 out of 100 adults have it [1]. AD is a skin disease that causes intense itching, leading to scratching. It can also result in dry skin and itchy rashes in specific areas, such as the folds of the arms and legs. In some cases, it can cause the thickening of the skin. AD is a condition in which macrophages, lymphocytes, and granulated mast cells release various chemical mediators and high levels of immunoglobulin (Ig) E in the serum, resulting in inflammation [2]. Furthermore, it has been reported that AD is provoked by the Th1/Th2 immune response. During the acute phase of AD, skin lesions are infiltrated by CD4^+^ T cells, which primarily secrete the Th2 cytokines IL-4, IL-5, and IL-13. However, in the chronic phase, Th1 cells secrete IFN-γ, leading to Th1-type inflammation and delayed-type hypersensitivity (DTH). This immune response leads to tissue remodeling and dermal thickening through the accumulation of collagen [3,4].

Although the cause of AD is not well known, it is presumed to be a result of a combination of environmental and genetic factors. Environmental factors, such as solar radiation and pollutants, can increase oxidative stress in the skin. Continuous exposure to environmental factors can cause skin inflammation that exceeds the skin’s antioxidant defense capacity. Thus, oxidative stress has been suggested to play a role in the development of AD [5,6]. Since increased oxidative stress caused by environmental factors is associated with AD, anti-oxidants are effective at treating AD in a therapy as they can reduce oxidative stress. It is widely recognized that antioxidants, such as coenzyme Q10 and poly phenolic compounds vitamin E and vitamin C, have the ability to protect the skin from damage caused by reactive oxygen species (ROS) [7,8].

Vitamin C is found in vegetables and fruits as a water-soluble compound. Vitamin C can reduce the number of unstable species of oxygen, nitrogen, and sulfur radicals. It has been widely used in the treatment and prevention of numerous diseases. Vitamin C plays a role in the process of skin cell development by enhancing the function and production of ceramide synthase [9]. Recent research has revealed that individuals suffering from AD exhibit reduced levels of vitamin C in the dermis and plasma [10]. In addition, vitamin C has the potential to enhance the management of chronic inflammation and have a beneficial impact on conditions such as AD and allergic diseases [11]. On the other hand, a number of studies have concluded that vitamin C does not have any significant impact on the process of allergic sensitization or the development of allergic rhinitis [12]. 

In order to achieve anti-allergic effects, it is necessary to supplement patients with a high dose of vitamin C to maintain the desired levels of vitamin C in various organs. However, vitamin C is easily oxidized, unstable, and decomposes into biologically inactive compounds when it is in contact with environmental factors such as pH, air, UV light, and temperature [13]. To enhance the bioavailability and stability of vitamin C, several kinds of formulations such as microparticles, nanoparticles, liposomes, and chemical modification have been reported [14,15].

Aptamers, which are RNA oligonucleotides or short single-stranded DNA molecules, have the ability to selectively bind to specific molecules. Aptamers offer numerous advantages due to their compact size, ability to be rapidly generated through chemical synthesis, adaptable chemical modifications, exceptional stability, and non-immunogenic properties. As previously stated, our team has successfully developed NXP081, a groundbreaking DNA aptamer that enhances the antioxidant power of vitamin C [16]. The objective of this study was to investigate the effects of NXP081 in a DNFB-induced AD model.

## 2. Materials and Methods

### 2.1. Animals

We obtained male Balb/c mice at 8 weeks of age from Japan’s SLC (Shizuoka, Japan) and maintained them in a controlled environment. The mice were kept under specific pathogen-free (SPF) conditions and were raised in an air-conditioned animal room (22 ± 2 °C; RH 50% ± 10%). They were granted unrestricted access to both water and food. All the experiments followed guidelines issued by Kyung Hee University’s animal Welfare ethics committee (KHUASP-18-112). All the experiments were conducted following the research guidelines set by the U.S. National Institutes of Health. The procedures strictly adhered to the recommended protocols, and they were followed meticulously to ensure compliance with the established guidelines.

### 2.2. NXP081 Preparation

The aptamer consists of oligonucleotides based on single-stranded DNA or RNA, which can selectively bind to the target molecule. They are isolated using the Systematic Evolution of Ligands by Exponential Enrichment (SELEX) method [16]. The purified DNA aptamer (Aptamin C132) obtained from Nexmos Co., Ltd. (Yongin-si, Republic of Korea), was dissolved in PBS at 95 °C for 5 min. It was then gradually cooled to room temperature, resulting in the formation of a tertiary structure. Aptamin C132 was mixed with vitamin C (Sigma-Aldrich, St. Louis, MO, USA) in PBS to produce a high dose of NXP081 (50 mg/kg of vitamin C and 5 mg/kg of Aptamin C132) and a low dose of NXP081 (5 mg/kg of vitamin C and 5 mg/kg of Aptamin C132).

### 2.3. DNFB-Induced AD

Dermatitis was triggered by the repetitive application of a solution containing 0.2% DNFB, with a precise volume of 25 μL. The solution was dissolved in a mixture of acetone and olive oil at a ratio of 3:1. This solution was carefully applied to both the external and internal surfaces of the mice’s ears. Additionally, on days 3 and 6, a total of 100 μL of the same solution was carefully applied to the shaved skin of the dorsal region. A total of 0.3% DNFB was applied to the skin surface of the ear and dorsal region on days 9, 12, 15, 18, 21, 24, and 27 to further challenge the DNFB-sensitized mice, and the control was treated with the same volumes of acetone and olive oil (3:1). From day 9 to day 27, vitamin C or Aptamin C132 or NXP081 was intraorally administrated daily. The thickness of the ear was measured using a thickness gauge (Digimatic Indicator, Mitutoyo, Tokyo) every other day.

### 2.4. Experimental Design

The animals were divided into six groups in a random manner: group 1; control, group 2; DNFB, group 3; DNFB + vitamin C (50 mg/kg), group 4; DNFB + Aptamin C132 (5 mg/kg), group 5; DNFB + low-dose NXP081; and group 6. DNFB + high-dose NXP081. In order to conduct the experiment, five animals were assigned to each group. The mice in the vitamin C, Aptamin C132, and NXP081 groups were orally administered injections daily, starting from day 9 and continuing until day 27 (Figure 1).

### 2.5. Histological Analysis

Twenty-four h after the last DNFB application, the dorsal skin was removed, and the ear samples were collected. These samples were then treated with paraffin and stained with hematoxylin and eosin (H&E) to histologically evaluate them. The mast cell count in the toluidine blue staining cross sections was determined by manually counting the cells under a microscope. The results are expressed as the number of cells per unit length of tissue section (400 µm × 400 µm/field). Representative images were captured using a camera attached to a microscope (Olympus, Tokyo, Japan) at 100× magnification.

### 2.6. IFN-γ Production and IL-4 Production Measurements

T lymphocytes were isolated from lymph nodes. CD4^+^ T cells were purified using a Biomag separation column (QIAGEN, Hilden, Germany) following the protocol provided by the manufacturer. Isolated CD4^+^ T cells (1 × 10^6^) were then transferred to model 3524 24-well flat-bottom culture plates (Corning, Corning, NY, USA) and cultured in RPMI-1640 medium supplemented with 10% heat-inactivated fetal bovine serum and 50 µM β-mercaptoethanol. The cells were then stimulated with 1 µg/mL of immobilized anti-CD3 (BD Pharmingen, San Diego, CA, USA) and 2 µg/mL of soluble anti-CD28 antibody for 50 h at 37 °C in a 5% CO_2_ atmosphere. After incubation, the supernatants of the culture were collected, and the concentrations of IFN-γ and IL-4 were measured using enzyme-linked immunosorbent assay (ELISA) (Biolegend, San Diego, CA, USA) kits.

### 2.7. Serum IgE Measurements

On the 27th day, DNFB was applied, and the serum samples were collected after 24 h. The concentration of IgE in the serum was measured using BD OptEIATM Set Mouse IgE ELISA kits (BD Pharmingen, San Diego, CA, USA).

### 2.8. Statistical Analysis

The outcome is indicated by the standard error of the mean, which is represented as ±(S.E.M.). The significance of the alteration was assessed using Student’s *t*-test. The utilization of variance analysis serves as a sophisticated and compelling method to assess the disparities among the experimental groups. Statistical significance was considered when the *p*-values were below 0.05.

## 3. Results

### 3.1. NXP081 Decreases DNFB-Induced AD-like Skin Inflammation

In order to assess the impact of NXP081 on AD-like skin lesions induced by DNFB, AD-like skin lesions were elicited via the repetitive sensitization of DNFB in Balb/c mice. As expected, ear thickness gradually increased with the amount of DNFB sensitization after day 10 (Figure 2). The treatment with vitamin C (50 mg/kg) or Aptamin C132 (5 mg/kg) caused a slightly larger decrease in ear thickness after day 22 compared to that caused by the DNFB treatment. Additionally, the treatment with NXP081 at low (5 mg/kg of vitamin C and 5 mg/kg of Aptamin C132) and high (50 mg/kg of vitamin C and 5 mg/kg of Aptamin C132) doses more significantly reduced the DNFB-induced ear thickness after day 22 compared with that caused by the DNFB treatment (Figure 2). Furthermore, we found similar effects of the NXP081 treatment on dorsal skin lesions (Figure 3A). The repeated application of DNFB paint can result in a significant influx of inflammatory cells into the dermal layer. The significant thickening of the epidermis was observed when examining dorsal skin lesions using H&E staining (Figure 3A). Even though the use of vitamin C or Aptamin C132 only did not lead to a decrease in DNFB-induced epidermal hyperplasia; however, the use of NXP081 resulted in a significant reduction in DNFB-induced epidermal hyperplasia. This reduction occurred in a dose-dependent manner (Figure 3A,B).

### 3.2. NXP081 Reduces Infiltration of Mast Cells in Skin Lesions

The migration of mast cells to epidermal layers is an important marker of AD [17]. Thus, we also investigated the migration pattern of mast cells after DNFB stimulation. The skin samples taken from the inflamed area showed a notable increase in the migration and degranulation of mast cells in the group that was treated with DNFB, as observed using toluidine blue staining (Figure 4A). The migration of mast cells was significantly reduced when treated with NXP081, and the extent of the reduction was dose-dependent (Figure 4A,B). However, the treatment with vitamin C or Aptamin C132 did not have any effect on mast cell migration (Figure 4A,B).

### 3.3. NXP081 Reduces the Levels of IFN-γ in Activated CD4^+^ T Cells

AD skin lesions display inflammation that is either dominated by Th1 or Th2 [18]. Therefore, experimental studies were conducted to determine if NXP081 plays a role in regulating the immune responses of CD4^+^ T cells in the lymph nodes, with a specific focus on Th1 and Th2 reactions. Firstly, we isolated CD4^+^ T cells from the lymph nodes. These cells were then stimulated with antibodies to CD3 and CD28. As anticipated, the levels of IFN-γ and IL-4 production demonstrated a significant increase following the administration of DNFB. On the other hand, the treatment of NXP081 or vitamin C suppressed the production of IFN-γ in activated CD4^+^ T cells, but did not affect IL-4 (Figure 5A,B). Such observations were confirmed several times, indicating that NXP081 is effective in suppressing the Th1 immune response involved in the AD pathomechanism. The serum IgE levels are also elevated in an animal model of AD [19]. We decided to determine if the NXP081 treatment would have an impact on the serum IgE levels. On day 28, blood samples were collected to measure the levels of IgE in the serum using ELISA. As anticipated, the levels of serum IgE demonstrated a significant increase following the administration of DNFB (Figure 5C). However, the NXP801 treatment showed a slightly larger decrease in the serum IgE level compared to that of the DNFB treatment group, although this difference was not statistically significant (Figure 5C).

## 4. Discussion

The pathogenesis of AD is not well known. It might be due to a combination of genetic and environmental factors. Environmental factors such as air pollution and chemical substances have been found to cause oxidative stress and disrupt the normal functioning of the skin barrier in humans. The are causes of AD symptoms [5,6]. Oxidative stress is a condition that occurs when the production of harmful oxidants within an organism’s cells exceeds its ability to neutralize them with antioxidants. This imbalance disrupts the delicate equilibrium necessary for the proper functioning of cells and can lead to various detrimental effects on the organism’s overall health and well-being. This imbalance results in an excess of oxidants, which can cause damage to the cells and tissues [2]. The skin plays a vital role in safeguarding the body against various external threats, including pollutants and allergens, by acting as a protective barrier. Therefore, the skin is a primary site for oxidative stress due to continuous ROS production within keratinocytes, which is triggered by both external and internal stimuli. Oxidative stress has been implicated in the development and progression of AD. Thus, antioxidants might be attractive therapeutic options. Despite the fact that vitamin C possesses strong antioxidant properties, its application is limited due to its susceptibility to oxidation. To overcome this oxidation problem, we developed a new substance, NXP081, a compound of Aptamin C132 and vitamin C, that can reduce the rate of vitamin C oxidation in aqueous solutions [16]. Several aptamers can also enhance the stability of vitamin C, increase skin moisturization, and improve pruritus [16]. They show neuroprotective effects in a neurotoxin 1-methyl-4-phenyl-1,2,3,6-tetrahydropyridine (MPTP)-induced Parkinson’s disease model [20]. In addition, they have the ability to reduce the inflammation caused by the house dust mite extract in keratinocytes [21].

In this present study, we examine to the impact of NXP081 on a DNFB-induced AD model. The anti-inflammatory properties of vitamin C have long been recognized for their ability to reduce the inflammation of the skin. As expected, the repetitive sensitization of DNFB increased the ear thickness from day 13 to day 25 during this experiment (Figure 2). Although the vitamin C or Aptamin C132 treatment slightly reduced the ear thickness of DNFB-sensitized mice (Figure 2), vitamin C or Aptamin C132 failed to decrease the level of the DNFB-induced epidermal thickness of the dorsal skin (Figure 3). In contrast, the NXP801 treatment significantly reduced the ear thickness and epidermal thickness of the dorsal skin (Figure 2). Mast cells play a crucial role in the development and progression of AD and various allergic conditions [17]. Several studies have shown that vitamin C administration can attenuate mast-cell-mediated bronchial hypersensitivity or allergies [22]. Furthermore, in cases of AD caused by house mites, the application of a topical ointment containing a combination of vitamin C and zinc oxide has been observed to decrease the presence of mast cells in the skin of mice that have been exposed to mites [23]. In a manner similar to the previously mentioned reports, the migration of mast cells was significantly reduced in the groups that received dose-dependent amounts of NXP081 (Figure 4A,B). Elevated IgE levels are commonly associated with allergies, making this an important factor to consider. The IgE antibody produced by B cells has the ability to activate mast cells; activation subsequently leads to the release of histamine and Th2 cytokines [24]. Noh et al. have reported that a vitamin C treatment can decrease IgE antibody response in vivo [25]. However, numerous research studies have consistently found that consuming vitamin C in the diet does not have a significant impact on the IgE levels of individuals diagnosed with AD or allergic rhinitis [12,26]. In the present study, the NXP081 treatment slightly, but not significantly, decreased the serum IgE level (Figure 5C). It is believed that this discrepancy arose from variations in the amount of vitamin C treatment and the experimental conditions. Additional research is required to confirm this hypothesis. In general, AD shows a Th1 or Th2 immune response depending on whether it is a case of acute or chronic inflammation, respectively. In the case of acute AD, there is a Th2 immune response. In the case of chronic AD, there is a Th1 immune response. In a study conducted by Noh et al., it was found that administering vitamin C can lead to an increase in the production of Th1 cytokines, such as IL-2, IFN-γ, and TNF-a, while simultaneously reducing the secretion of Th2 cytokine IL-4 in a response known as DTH [25]. On the other hand, there have been numerous conflicting findings regarding the impact of vitamin C on the immune response [12,26]. There have been reports of unaffected humoral responses in both humans and Balb/c mice [27,28]. In this study, the NXP081 treatment decreased the production of IFN-γ, but not IL-4. Therefore, the suppression of immune response caused by the NXP081 treatment is more sensitive to the Th1 immune response. During the study, we observed that the low-dose NXP081 group exhibited a more significant decrease in the secretion of IFN-γ in the activated CD4^+^ T cells and decreased ear thickness in the mice compared to those of the vitamin C group. It is worth noting that the low-dose NXP081 group, which were given 5 mg/kg vitamin C, showcased this effect despite receiving a lower dosage of vitamin C than the vitamin C group did, which were given 50 mg/kg (Figure 4). Chiu et al. have reported that DNA aptamer can slow the oxidation rate of vitamin C in aqueous solutions [16]. Thus, the greater effect in the low-dose NXP081 treatment group might be due to the slower oxidation of vitamin C due to Aptamin C132. However, more experiments are needed to clarify this. The present study reports that NXP081 could be a candidate for AD therapy by prolonging vitamin C functions.

## Figures and Tables

**Figure 1 nutrients-15-04172-f001:**
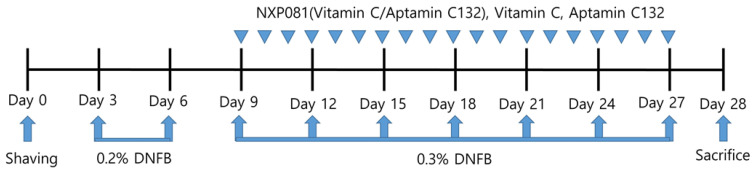
Schedule of DNFB-induced AD mice model.

**Figure 2 nutrients-15-04172-f002:**
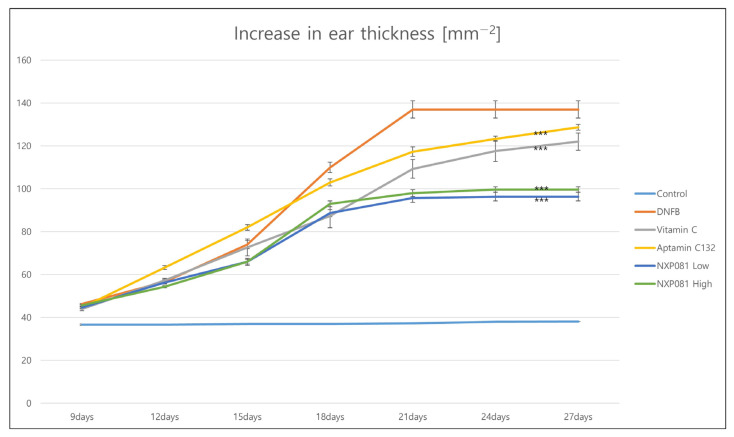
The impact of NXP081 treatment on ear skin inflammation was assessed using a Balb/c mouse model of AD induced by DNFB. The thickness of the ears was measured every three days using a thickness gauge to quantify the degree of swelling. NXP081 was orally administered daily from 9 days after DNFB sensitization. The values are presented as the mean ± SEM of five mice per group. *** *p* < 0.0001.

**Figure 3 nutrients-15-04172-f003:**
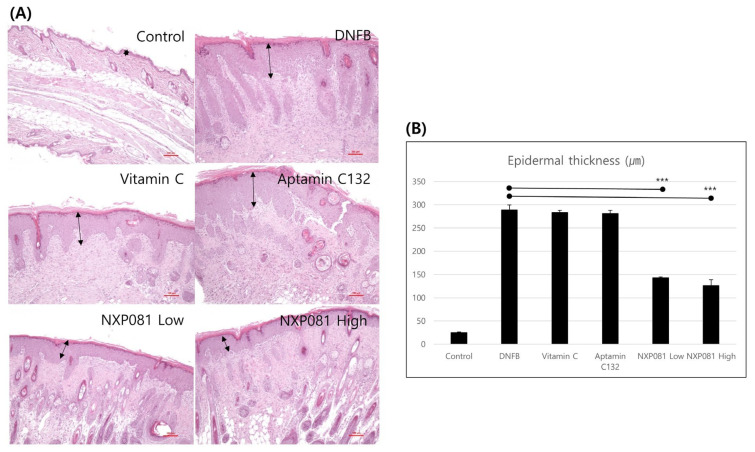
The impact of NXP081 treatment on dorsal skin inflammation was assessed using a Balb/c mouse model of AD induced by DNFB. (**A**) Dorsal skin samples were taken and subjected to staining with H&E (scale bar = 100 µm). (**B**) This bar is the average of the epidermal thickness of the three tissues. Data are presented as the mean ± SEM. ***, *p* < 0.0001.

**Figure 4 nutrients-15-04172-f004:**
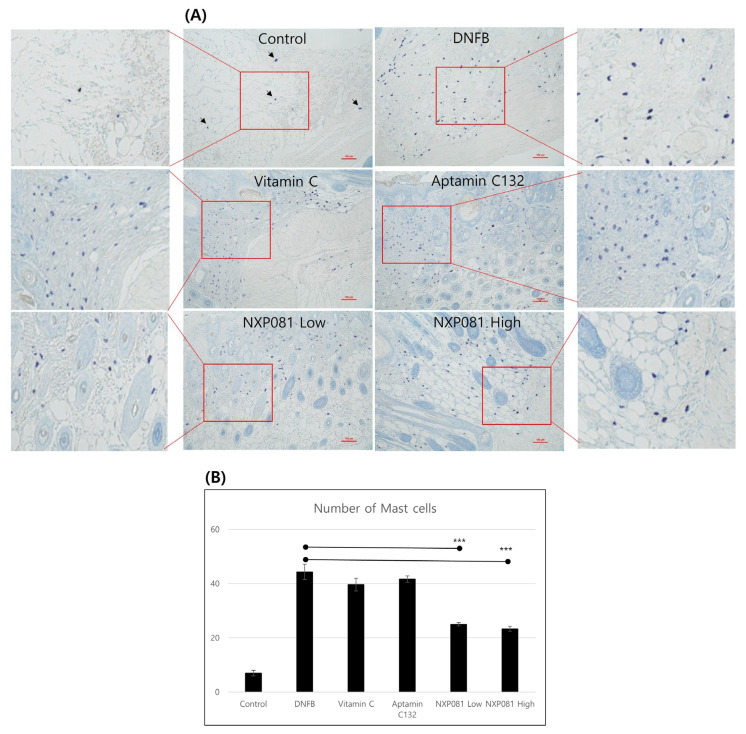
The impact of NXP081 on the quantity of mast cells in the dorsal skin of DNFB-stimulated Balb/c mouse model. (**A**) The histological evaluation was conducted using toluidine blue staining, which resulted in the presence of cells exhibiting a vivid purple color (arrows) (scale bar = 100 µm). (**B**) The number of mast cells in AD. The number of cells was counted for each section using 3 randomly selected fields (400 µm × 400 µm/field). Data are presented as the mean values ± SEM. ***, *p* < 0.0001.

**Figure 5 nutrients-15-04172-f005:**
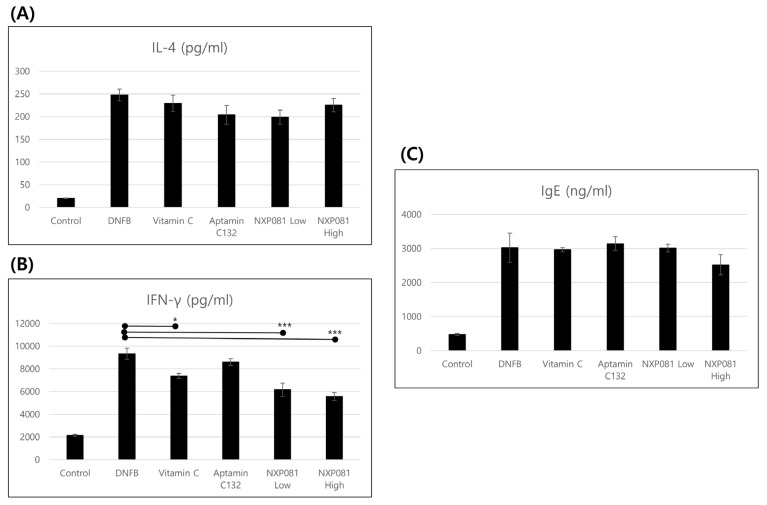
The impact of NXP081 on the production of IL-4 (**A**) and IFN-γ (**B**) by activated CD4^+^ T cells and serum IgE (**C**) level in DNFB-stimulated Balb/c mice. The lymph nodes were removed 24 h after the final administration of DNFB and purified CD4^+^ T cells using a Biomag separation column. CD4^+^ T cells (1 × 10^6^ cells/mL) were stimulated for 48 h with anti-CD28 antibody (2 µg/mL) and anti-CD3 antibody (1 µg/mL) on a plate. The levels of IL-4, IFN-γ, and IgE were determined using ELISA with four samples per group. Data are presented as the mean values ± SEM. * *p* < 0.05; *** *p* < 0.0001.

## Data Availability

Not applicable.

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
