# Peer review of "NXP081, DNA Aptamer–Vitamin C Complex Ameliorates DNFB-Induced Atopic Dermatitis in Balb/c Mice"

_nutrients, 2023, doi:10.3390/nu15194172_

Round 1
Reviewer 1 Report
Review comments of manuscript Nutrients-2608827 entitled:
NXP081, DNA aptamer-Vitamin C complex ameliorates DNFB-induced atopic dermatitis in Balb/c mice
The authors in the manuscript developed NXP081, a single-stranded DNA aptamer that can bind to Vitamin C with specificity, thus reducing Vitamin C oxidation. They investigated the effect of NXP081 in a 2,4-dinitrofluorobenzene(DNFB)-induced atopic dermatitis animal model. The results demonstrated that NXP081 significantly lowered ear swellings and improved skin lesions induced by DNFB, reduced infiltration of mast cells into DNFB-treated skin lesions, and inhibited interferon-gamma (IFN-γ) production in activated CD4+ T cells from draining lymph nodes. The finding provides potential use for NXP081 in treating atopic dermatitis.
And that I would like to address the points as follows:
1. In the preparation of NXP081, line 86-94, Aptamin C132 was mixed with Vitamin C. However, the description did not prove the structure characterization. It is suggested that more information should be provided, such as Changes in spectral characteristics.
2. In the administration to animals, the detailed drug solution preparation of Vitamin C, Aptamin C132 and NXP081 was missing, what solvent, concentration, etc.
3. Vitamin C or Aptamin C132 failed to decrease the level of DNFB-induced epidermal thickness of back skin (Fig. 2), lines 209-210. There was no in Fig. 2.
4. The results are expressed in the statistic as means ± standard error of the mean (S.E.M.), lines 124-128. Please check and confirm if all data's standard errors are expressed as SEM, no SD?
Minor editing of English language required.
Author Response
Thank you very much for your kind and important comments.
- In the preparation of NXP081, line 86-94, Aptamin C132 was mixed with Vitamin C. However, the description did not prove the structure characterization. It is suggested that more information should be provided, such as Changes in spectral characteristics.
- Your comment is a very important part, but in this paper, we received the produced aptamer from Nexmos Co, which is a collaborative research, and details on the spectrum and structure of these information are in refenece 16 related to Nexmos Co.
- Chiu, A.S.; Sankarapani, V.; Drabek, R.; Jackson, G.W.; Batchelor, R.H.; Kim, Y-S. Inhibition of Vitamin C oxidation by DNA aptamers. Aptamers. 2018, 2, 28-35.
- In the administration to animals, the detailed drug solution preparation of Vitamin C, Aptamin C132 and NXP081 was missing, what solvent, concentration, etc.
- Vitamin C, Aptamin C132, and NXP081 were dissolved in PBS, and the concentration was added to materials and methods as follows.
“The purified DNA aptamer (Aptamin C132), obtained from Nexmos Co., Ltd., was dis-solved in PBS at 95°C for 5 minutes. It was then gradually cooled to room temperature, resulting in the formation of a tertiary structure. Aptamin C132 was mixed with vitamin C (Sigma-Aldrich, MO, USA) in PBS to produce a high dose of NXP081 (50 mg/kg of vitamin C and 5 mg/kg of Aptamin C132) and a low dose of NXP081 (5 mg/kg of vitamin C and 5 mg/kg of Aptamin C132).”
- Vitamin C or Aptamin C132 failed to decrease the level of DNFB-induced epidermal thickness of back skin (Fig. 2), lines 209-210. There was no in Fig. 2.
- Thank you very much for your accurate point. As you pointed out, I revised the contents as follows.
‘Although vitamin C or Aptamin C132 treatment slightly reduced ear thickness of DNFB sensitized mice (Fig. 2), vitamin C or Aptamin C132 failed to decrease the level of DNFB induced epidermal thickness of dorsal skin (Fig. 3).”
- The results are expressed in the statistic as means ± standard error of the mean (S.E.M.), lines 124-128. Please check and confirm if all data's standard errors are expressed as SEM, no SD?
- Thank you very much for your accurate point. As you pointed out, I checked that it is SEM. Our data is SEM.
Reviewer 2 Report
This article investigates the effect of NXP081 on the animal model of DNFB induced atopic dermatitis. The results showed that oral administration of NXP081 significantly reduced ear swelling caused by DNFB and improved skin damage. NXP081 significantly reduced the infiltration of mast cells and the production of IFN-γ in CD4+ T cells. But there are some issues:
1. What is the mechanism of NXP081 preparation, i.e., how does Aptamin C132 bind to vitamin C?
2. Is it reasonable to take NXP081 orally? Is there literature on aptamers being able to be taken orally?
3. In Figure 2, Aptamin C132 has significant anti-inflammatory effects compared to the model group. Does Aptamin C132 have physiological activity?
4. The experimental animals are divided into 6 groups, which contradict the article's statement of 4 groups.
5. The order of many figures (A, B, C, etc.) needs to be indicated, and figures 4F and 4G cannot be found.
6. Vitamin C significantly reduces IFN-γ. Why does it have no impact on other indicators?
7. The introduction mentions "a high level of immunoglobulin (Ig) E in the serum" and "Th2 type inflammation causes infiltration of CD4+T cells which secrete interleukin (IL)-4, IL-5, and IL-13". Why does NXP081 not have a significant impact on IgE and IL-4?
Author Response
Thank you very much for your kind and important comments.
- What is the mechanism of NXP081 preparation, i.e., how does Aptamin C132 bind to vitamin C?
- Appaminc132 is DNA in the DNA library that binds to vitamin C and bovine serum albumin. A specific method is found in Reference No. 16.
- Chiu, A.S.; Sankarapani, V.; Drabek, R.; Jackson, G.W.; Batchelor, R.H.; Kim, Y-S. Inhibition of Vitamin C oxidation by DNA aptamers. Aptamers. 2018, 2, 28-35.
- Is it reasonable to take NXP081 orally? Is there literature on aptamers being able to be taken orally?
- Thank you for the important point.
We did peritoneal administration, oral administration, and skin application as a preliminary experiment. Among them, oral administration was the best. And a Nexmos, collaborate company, is planning to develop a drink in the future, so we experimented with oral administration
- In Figure 2, Aptamin C132 has significant anti-inflammatory effects compared to the model group. Does Aptamin C132 have physiological activity?
- Thank you for your exact point. It is known that some bacterial DNA stimulates immune cells to induce an immune response, but it is difficult to give an exact answer to whether this Aptamer C132 induces an immune response.
- The experimental animals are divided into 6 groups, which contradict the article's statement of 4 groups.
-Thank you for your exact point. I corrected it as below.
“The animals were divided into six groups in a random manner: group 1. Control, group 2. DNFB, group 3. DNFB + vitamin C (50 mg/kg), group 4. DNFB + Aptamin C132 (5 mg/kg), group 5. DNFB + low dose NXP081 and group 6. DNFB + high dose NXP081.”
- The order of many figures (A, B, C, etc.) needs to be indicated, and figures 4F and 4G cannot be found.
- Thank you for your exact point. I corrected it
- Vitamin C significantly reduces IFN-γ. Why does it have no impact on other indicators?
- The amount of IFN-g produced when stimulating immune cells is significantly higher than that of IL-4. Therefore, I think it would have been easy to observe the changes depending on the vitamin C treatment. And I think the results will be different depending on the experimental conditions because vitamin C is easily oxidized and loses its function. We found that NXP081(Aptamer C132+vitamin C) treatment was very effective in preventing oxidation of vitamin C.
- The introduction mentions "a high level of immunoglobulin (Ig) E in the serum" and "Th2 type inflammation causes infiltration of CD4+T cells which secrete interleukin (IL)-4, IL-5, and IL-13". Why does NXP081 not have a significant impact on IgE and IL-4?
- Thank you for your exact point. There are many different results from experiments on vitamin C. Some have reported that Th1 and 2 types of immune responses are also reducedb, but other conditions of experiment do not affect Th2 type immune responsesa. This depends on the experimental conditions (treatment capacity, stimulation conditions, etc.). I think our experiment is reproducible because the Th2 type immune response has not changed and there has been no IgE change in conjunction with it.
- Lee, S.; Ahn, K.; Paik, H.Y.; Chung, S-J. Serum immunoglobulin E levels and dietary intake of Korean infants and young children with atopic dermatitis. Nutr Res Pract. 2012, 6(5), 429-435.
- Noh, K.; Lim, H.; Moon, S-K.; Kang J.S.; Lee, W.J.; Lee, D.; Hwang, Y.I. Mega-dose Vitamin C modulates T cell functions in Balb/c mice only when administered during T cell activation. Immunol Lett. 2005, 98, 63-72.
Reviewer 3 Report
To authors
The authors have investigated the effect of NXP081, (a vitamin C, aptamer combination) in a 2,4-dinitrofluorobenzene(DNFB)-induced atopic dermatitis animal model with respect to reducing oxidative damage. The concluded that oral administration of NXP081 significantly lowered ear swellings and improved skin lesion induced by DNFB. Additionally, treatment with NXP081 significantly reduced infiltration of mast cells into DNFB-treated skin lesions and inhibited interferon-gamma (IFN-γ) production in activated CD4+ T cells from draining lymph nodes.
The paper is well written with no language concerns and easy to follow. I do have the following recommendations to make it a stronger paper.
Abstract
Line 17 – The chemical name is a different color than the rest of the text.
Introduction
Around lines 33-35 – You could mention IFN-gamma in the intro as this is the cytokine that you are investigating.
Line 61 – Aptamer should be plural -“Aptamers”
Material and Methods
Indicate what type of food they were fed and whether it contains vitamin C
I would place the “NXP081 preparation” paragraph before the “DNFB-induced dermatitis paragraph” so that the Aptamin C132 or NXP081 terms are explained first.
Results
Section 3.1 I would recommend including the dosages here as well, not just in the material and methods section
Figure 3 should have the letters “A” and “B” labels on the Figures as well to match what is in the figure description.
Figure 3A “control" panel looks like a different magnification than the other photos based on the dermis and subcutaneous tissue even thought the magnification bar does not indicate it.
The magnification bar is too small to see even with figure magnification. The magnification should be indicated in the figure description as well.
The bar graphs in Figure 3B do not correspond well to the Figure images in 3A (Ex NXP081 looks more than 4 times the control in the photos) which suggest issues with the control photo.
Figure 4 should have the letters “A” and “B” labels on the Figures as well to match what is in the figure description. Or labeled up to letter G as referred to in the discussion
Magnification indicator in figure 4 A is too small to see without significant magnification of the figure so it should be indicated in the figure description as well.
Cells are difficult to see without magnification of the figure.
In figure 4 B, indicate the unit area (Example “Number of mast cells/field”)
Figure 5 requires A, B, C labels as well.
In Figure 5 description, indicate how many samples were used to establish the statistics as it is not indicated here or in the materials and methods.
Discussion
Mention whether there is any significance to mice synthesizing their own vitamin C.
Indicate why a dermal preparation/cream, containing vitamin C was not considered for a skin condition. One was noted in the discussion.
What you were testing was more the effect of vitamin C to inhibit the worsening of AD rather than a treatment as it was attempted during the development of the chemically induced AD. Nothing wrong there, just be aware, this was the test conditions.
Line 203 – A new paragraph could be started at “In the present study,….”
English quality is excellent, only a couple of typos but would recommend the use of paragraphs in the discussion section.
Author Response
Thank you very much for your kind and important comments.
Abstract
Line 17 – The chemical name is a different color than the rest of the text.
- Thank you for your comment. I corrected it.
Introduction
Around lines 33-35 – You could mention IFN-gamma in the intro as this is the cytokine that you are investigating.
- Thank you for your kind comment, I corrected it as follows.
“However, in the chronic phase, Th1 cells secrete IFN-γ, leading to Th1-type inflammation and delayed-type hypersensitivity (DTH). This immune response leads to tissue remodeling and dermal thickening through the accumulation of collagen [3,4].”
Line 61 – Aptamer should be plural -“Aptamers”
- Thank you for your comment. I corrected it.
Material and Methods
Indicate what type of food they were fed and whether it contains vitamin C
- Thank you for your exact point. I fed mice normal food without vitamin C.
“They were granted unrestricted access to both water and food.”
I would place the “NXP081 preparation” paragraph before the “DNFB-induced dermatitis paragraph” so that the Aptamin C132 or NXP081 terms are explained first.
- Thank you for your comment. I corrected it.
Results
Section 3.1 I would recommend including the dosages here as well, not just in the material and methods section
- Thank you for your critical comment! As you suggested, I corrected it.
‘Treatment with vitamin C (50 mg/kg) or Aptamin C132 (5 mg/kg) slightly decreased ear thickness after day 22 compared to DNFB treatment. Additionally, treatment with NXP081 at low (5 mg/kg of vitamin C and 5 mg/kg of Aptamin C132) and high (50 mg/kg of vitamin C and 5 mg/kg of Aptamin C132) doses significantly reduced DNFB-induced ear thickness after day 22 compared with DNFB treatment”
Figure 3 should have the letters “A” and “B” labels on the Figures as well to match what is in the figure description.
- Thank you for your critical comment! As you suggested, I corrected it.
Figure 3A “control" panel looks like a different magnification than the other photos based on the dermis and subcutaneous tissue even thought the magnification bar does not indicate it.
- Thank you for your exact point. As pointed out, I corrected the figure to re-match the scale bar.
The magnification bar is too small to see even with figure magnification. The magnification should be indicated in the figure description as well.
- Thank you for your exact point. As pointed out, I corrected the figure and legend.
The bar graphs in Figure 3B do not correspond well to the Figure images in 3A (Ex NXP081 looks more than 4 times the control in the photos) which suggest issues with the control photo.
- Thank you for your exact point. As pointed out, I corrected the picture to re-match the scale bar.
Figure 4 should have the letters “A” and “B” labels on the Figures as well to match what is in the figure description. Or labeled up to letter G as referred to in the discussion
- Thank you for your critical comment! As you suggested, I corrected it.
Magnification indicator in figure 4 A is too small to see without significant magnification of the figure so it should be indicated in the figure description as well.
- Thank you for your critical comment! As you suggested, I corrected it.
Cells are difficult to see without magnification of the figure.
- Thank you for your critical comment! As you suggested, I amplified figure.
In figure 4 B, indicate the unit area (Example “Number of mast cells/field”)
- Thank you for your exact point. As pointed out, I corrected the figure and legend.
“The number of cells was counted for each section using 3 randomly selected fields (400 µm x 400 µm/field).”
Figure 5 requires A, B, C labels as well.
- Thank you for your exact point. ,I corrected it.
In Figure 5 description, indicate how many samples were used to establish the statistics as it is not indicated here or in the materials and methods.
- Thank you for your critical comment! As you suggested, I corrected it.
- . “The levels of IL-4, IFN-γ, and IgE were determined using ELISA with four samples per group. Data are presented as the mean values ± SEM. ***, p < 0.0001.
Discussion
Mention whether there is any significance to mice synthesizing their own vitamin C.
-I'm sorry, but I didn't have many reference for vitamin C produced by mice, so I couldn't discussed the meaning of it.
Indicate why a dermal preparation/cream, containing vitamin C was not considered for a skin condition. One was noted in the discussion.
- Thank you for the important point.
We did peritoneal administration, oral administration, and skin application as a preliminary experiment. Among them, oral administration was the best. And a Nexmos, collaborate company, is planning to develop a drink in the future, so we experimented with oral administration
What you were testing was more the effect of vitamin C to inhibit the worsening of AD rather than a treatment as it was attempted during the development of the chemically induced AD. Nothing wrong there, just be aware, this was the test conditions._
- Sorry, I couldn’t understand it!
Line 203 – A new paragraph could be started at “In the present study,….”
- Thank you for your exact point. I corrected it.
Reviewer 4 Report
Overall, this manuscript reports evidence that stabilizing Vitamin C using DNA aptamer technology can significantly reduce the inflammatory response, mainly Th1 immune response, induced by DFNB treatment in mice. The study design and data reported were adequate to support the conclusions. Although, since this is an oral treatment that would be intended for AD disease in human, it would be important to understand how this would translate in human, which are much larger than a mouse, considering that 5 and 50mg of vitamin C was used in this study. This should be added to the discussion. Here few more suggestions:
Line 11: ... characterized by dry and itchy skin. (replace comma by "and")
Line 200: remove "an" after [16]
Line 225: replace "to solve this issue" by "to confirm this hypothesis"
The introduction and mainly the discussion would benefit if a scientific/medical writer would review and edit the manuscript. The proper information is there, but the flow could be improve, especially by combining same message/idea into one sentence instead of 2-3 sentences.
Author Response
Overall, this manuscript reports evidence that stabilizing Vitamin C using DNA aptamer technology can significantly reduce the inflammatory response, mainly Th1 immune response, induced by DFNB treatment in mice. The study design and data reported were adequate to support the conclusions. Although, since this is an oral treatment that would be intended for AD disease in human, it would be important to understand how this would translate in human, which are much larger than a mouse, considering that 5 and 50mg of vitamin C was used in this study. This should be added to the discussion. Here few more suggestions:
Thank you very much for your kind and important comments.
- In the study of natural products, two concentrations (5,50 mg/kg) were used because the most commonly used concentration was 1-100 mg/kg and the 50 mg/kg concentration was generally used in humans.
Line 11: ... characterized by dry and itchy skin. (replace comma by "and")
- Thank you for your critical comment! As you suggested, I corrected it.
“Atopic dermatitis (AD) is a chronic inflammatory disease characterized by dry and itchy skin.”
Line 200: remove "an" after [16]
- Thank you for your critical comment! As you suggested, I corrected it.
Line 225: replace "to solve this issue" by "to confirm this hypothesis"
- Thank you for your critical comment! As you suggested, I corrected it.
“Additional research is required to confirm this hypothesis.”
Round 2
Reviewer 1 Report
In the preparation of NXP081, lines 86-94, Aptamin C132 was mixed with Vitamin C. However, the description did not prove the structure characterization. It is suggested that more information should be provided, such as Changes in spectral characteristics.
The author explained that Nexmos Co provided NXP081, and details on the spectrum and structure of this information are in reference 16 (Chiu, A.S.; Sankarapani, V.; Drabek, R.; Jackson, G.W.; Batchelor, R.H.; Kim, Y-S. Inhibition of Vitamin C oxidation by DNA aptamers. Aptamers. 2018, 2, 28-35.)
Due to the difference in each experimental environment and operation, the type and yield of chemical reaction products are changed. Therefore, it is suggested to provide information on the chromatography, spectrum and mass spectrometry of NXP081 used in this experiment.
Author Response
Thank you for the important comment.
In our lab, rather than developing aptamer, we did functional research (in atopic dermatitis animal models). I am not an expert in the production of aptamer. As mentioned earlier, experiments on the physical properties and separation of aptamer for vitamin C are in Reference 16, and Dr. Kim established the experimental conditions and production. And because NXP081 supplied to us was used in the experiment in combination of aptamin C132 and vitamin C according to Nexmos' protocol, we did not confirm the physical properties of NXP081. I think your comment will be important when we commercialize it in the future and manage the quality of NXP081, which is stable. Thank you again for the important comment.
Reviewer 2 Report
The author has answered and revised the questions I raised. Question 2 requires further explanation. Appaminc132 is DNA in the DNA library that binds to vitamin C and bovine serum albumin. How does Appaminc132 maintain its stability during oral absorption?
Author Response
Thank you very much for your important comment.
As you pointed out, it is very important that aptamin C maintains the stability of vitamin C during oral intake. The specific mechanisms are currently unknown, but indirectly, Dr. Kim's previous research suggests that this structural combination (aptamer-vitamin C) has increased the antioxidant half-life of vitamin C by 1.7 times (ref. 16). I think aptamer will probably prevent the attack on the antioxidant site of vitamin C by free radicals. I think further research on this will be needed in the future, and thank you again for the important comments.
- Chiu, A.S.; Sankarapani, V.; Drabek, R.; Jackson, G.W.; Batchelor, R.H.; Kim, Y-S. Inhibition of Vitamin C oxidation by DNA aptamers. 2018, 2, 28-35.
Reviewer 3 Report
Thank you for addressing the majority on my comments/recommendations
Author Response
Thank you your kind comments.